# Common and Differential Traits of the Membrane Lipidome of Colon Cancer Cell Lines and Their Secreted Vesicles: Impact on Studies Using Cell Lines

**DOI:** 10.3390/cancers12051293

**Published:** 2020-05-20

**Authors:** Joan Bestard-Escalas, Albert Maimó-Barceló, Daniel H. Lopez, Rebeca Reigada, Francisca Guardiola-Serrano, José Ramos-Vivas, Thorsten Hornemann, Toshiro Okazaki, Gwendolyn Barceló-Coblijn

**Affiliations:** 1Lipids in Human Pathology, Health Research Institute of the Balearic Islands (IdISBa), Research Unit, University Hospital Son Espases, 07120 Palma, Spain; joanbe88@gmail.com (J.B.-E.); albert.maimo@ssib.es (A.M.-B.); danielhoracio.lopezlopez@ssib.es (D.H.L.); rebeca.reigada@ssib.es (R.R.); 2Department of Biology, University of the Balearic Islands, 07120 Palma, Spain; franciscaguardiola@hotmail.com; 3Valdecilla Research Institute (IDIVAL ), 39011 Santander, Spain; jvivas@idival.org; 4Microbiology Unit, University Hospital Marqués de Valdecilla, 39008 Santander, Spain; 5Spanish Network for Research in Infectious Diseases (REIPI), Institute of Health Carlos III (ISCIII), 28029 Madrid, Spain; 6Institute of Clinical Chemistry, University Hospital Zurich, University of Zurich, 8091 Zurich, Switzerland; thorsten.hornemann@usz.ch; 7Department of Hematology/Immunity, Kanazawa Medical University, Uchinada-machi, Kahoku-gun, Ishikawa 920-0293, Japan; dkbki308@kyoto.zaq.ne.jp

**Keywords:** colorectal cancer, lipidomics, lipid biomarkers, extracellular vesicles, cell lines, plasmalogens)

## Abstract

Colorectal cancer (CRC) is the fourth leading cause of cancer death in the world. Despite the screening programs, its incidence in the population below the 50s is increasing. Therefore, new stratification protocols based on multiparametric approaches are highly needed. In this scenario, the lipidome is emerging as a powerful tool to classify tumors, including CRC, wherein it has proven to be highly sensitive to cell malignization. Hence, the possibility to describe the lipidome at the level of lipid species has renewed the interest to investigate the role of specific lipid species in pathologic mechanisms, being commercial cell lines, a model still heavily used for this purpose. Herein, we characterize the membrane lipidome of five commercial colon cell lines and their extracellular vesicles (EVs). The results demonstrate that both cell and EVs lipidome was able to segregate cells according to their malignancy. Furthermore, all CRC lines shared a specific and strikingly homogenous impact on ether lipid species. Finally, this study also cautions about the need of being aware of the singularities of each cell line at the level of lipid species. Altogether, this study firmly lays the groundwork of using the lipidome as a solid source of tumor biomarkers.

## 1. Introduction

Colorectal cancer (CRC) is the third most commonly diagnosed malignancy and the fourth leading cause of cancer death in the world [1]. In some high-income countries, the implementation of screening programs has led to a significant reduction in CRC incidence in the population aged 50 years and over. Unexpectedly, the incidence in the population below this age has increased significantly for causes as yet unknown [2]. Therefore, the development of better tools for an accurate stratification of CRC patients is still highly needed. The multiple phenotypic and genetic particularities of tumors, as well as interindividual differences, are some of the difficulties hampering the development of cancer stratification tools. Currently, some studies propose classifying CRC patients according to methods based on multiple features, including histological, genetic, and epigenetic features [3]. In this scenario, the lipidome is emerging as a powerful tool to identify disease biomarkers [4,5,6,7,8], and increasing interest in the lipid metabolism is reflected in the sharp rise in the number of publications on this topic (from 20,000 in 1980 to 60,000 in 2018, according to Web of Science). 

Solid advances in the field of lipid analysis have facilitated access to detailed descriptions of many different lipidomes [4,7,9,10], while the irruption of imaging mass spectrometry techniques providing the bidimensional distribution of each lipid species across a tissue section has undoubtedly demonstrated how specific this distribution is [8,11,12,13,14]. Lipidomic results often yield complex scenarios, with hundreds of different lipids changing in a highly ordered and orchestrated manner by mechanisms yet to be defined. Consistently, the lipidome has proven to be highly sensitive not only to cell type but also to any biological process including differentiation, malignization, and death. In this sense, our previous study analyzing the changes in the lipid signature along the colon epithelium demonstrates that there is a strict regulation at the molecular species level, concomitant to the colonocyte differentiation process and that this regulation is clearly altered in the malignant tissue [8,15].

There is no doubt that the possibility of building complex profiles based on the combination of different molecular species, all measured at once, offers a wide variety of potential biomarkers for diagnosis. Furthermore, because of their tight participation in biological processes, membrane lipids could also be used to monitor the progression of a disease. In an effort to search for new non-invasive cancer biomarkers, the composition of extracellular vesicles (EVs) arises as a promising source of biomarkers [16,17]. Although in terms of lipid composition several studies have described the lipidome of EVs isolated from a diversity of biological sources [18,19,20,21,22,23,24], only a few of them explored how lipid molecular species of EVs could be used as biomarkers [25,26,27].

Altogether, the possibility of describing lipids at the level of molecular species has led to regaining the interest in investigating their specific role using study models, one of the most commonly used of which are commercial cell lines. Herein, we investigated whether the lipid fingerprint, of both cells and their derived EVs, could be used to distinguish cancer cell lines according to their degree of aggressiveness. Thus, we analyzed the lipidome of five human colon commercial cells, in particular, one healthy primary and four cancer cell lines, all broadly used in cancer research as a model of colorectal adenocarcinoma: HT29, SW480, and LS174t, isolated from primary tumors, and Colo 201, isolated from metastatic sites. The results demonstrate that the cell lipidome was indeed able to separate primary, in situ, and highly metastatic cell lines either by their cell lipid content or by their EV lipid composition. Furthermore, consistently with the literature, a profound impact on ethanolamine containing glycerophospholipids was observed in all cancer cells. Interestingly, a closer look into the changes in molecular species distribution within each lipid class revealed a striking shift of species from diacylphosphatidylcholine and diacylphosphatidyl-ethanolamine (PE) to PE plasmalogen. Unexpectedly, this distribution turned out to be highly selective for species presenting a saturated fatty acid at the sn-1 position and arachidonic (AA) or docosahexaenoic acid (DHA) at the sn-2 position. 

Altogether, these results demonstrate the capacity of lipid profiles, whether from cells or EVs, for sensing a wide range of physiological alterations and, consequently, in providing potential lipid biomarkers. This study also brings to the surface an alteration common to all the cancer cells analyzed, which involved a profound alteration in PE plasmalogen metabolism. Importantly, these changes go beyond a mere increase in mass, to affecting a very specific set of molecular species. Nevertheless, the heterogeneity within the analyzed cell lines does caution using a limited number of cell lines to study the role of particular molecular species in biological processes or to establish new biomarkers.

## 2. Results

### 2.1. Identification of Potential Lipid Markers for Cellular Malignization in Commercial Cell Line

Given the sensitivity of the lipidome to culture conditions, especially to high confluence conditions[28], all cell lines were characterized in terms of growth before starting the study. As a criterion, we prioritized that at the time of sample collection, cell confluence was similar in all cell lines. For this reason, healthy colon primary cells were plated at 3 × 10^4^ cells/cm^2^ for 48h, HT29 and LS174t cells were plated at 3 × 10^4^ cells/cm^2^ for 48h, SW480 cells were plated at 2 × 10^4^ cells/cm^2^ for 48h, and Colo 201 cells were plated at 2 × 10^4^ cells/cm^2^ for 72h. We used a protocol based on differential centrifugation and filtration processes to obtain an EV fraction highly-enriched in exosomes [29]. Although this fraction tested positive for exosome markers (CD9 antigen, 25 kDa), for simplicity it will be referred to as EVs. To assess the differences between tumor and healthy colon cells, we compared the lipidome obtained from a healthy primary cell line and four different CRC commercial cell lines. Further, we paid particular attention to the differences in lipid composition between cells isolated from a primary tumor (HT29, SW480, and LS174t) and those from a metastatic site (Colo 201) as they could identify as potential biomarkers for the metastatic process. 

The results of the lipidomic analysis of the five cell lines provided a rather complex scenario (Figure 1, Figure 2 and Figure 3). The PCA using cell membrane lipid content data, including the main phospholipids (PC, PE, PS, and PI) and sphingolipids (SM and Cer), showed that membrane lipid classes were able to separate primary from cancer cells (Figure 1A). The separation was due to changes in PE levels and, to a lesser degree, in PE plasmalogens and PC levels (Figure 1B). Consistently, ANOVA analysis confirmed the profound divergences in cell lipid composition between cell lines (Figure 1C and Appendix A). The most robust differences between primary and all cancer cells affected diacyl- and vinyl ether ethanolamine (PE and PE plasmalogens) levels. In tumor cells, PE levels decreased compared to primary cells (25.1 and 10.6%, respectively), while PE plasmalogen levels increased (5.4 and 15.2%, respectively). Finally, all cancer cell lines showed increased PC levels (44.9 vs. 51.7% primary vs. mean value of all tumor cells) and, consistent with previous studies [30,31], decreased sphingomyelin (SM) levels, which were significant in three of the four tumor cells (all but LS174t, 11.1 vs. 7.5%). 

To delve into these differences, a PCA was performed using all molecular lipid species detected (Figure 2). The results confirmed the capacity of the whole lipidome to separate the cell lines into three groups according to their malignancy; that is, primary cells (Prim) from in situ (HT29, SW480, and LS174t) and from highly metastatic cancer cells (Colo 201) (Figure 2A). Higher levels in PI38:3, SMd18:1/24:1, and Cerd18:1/24:1, and lower levels in PE P-16:0/22:6 and SMd18:1/16:0 accounted for the separation of the primary cells (Figure 2B). Colo 201 were separated from the in situ cells because of the higher content in PS and PE36:1, SMd18:1/16:0, and Cer18:1/24:0, and the lower content in Cer18:1/16:0 and 18:1/24:1 and PE P-16:0/20:4. Despite the fact that PCA was able to discriminate between the cell lines, it barely explained 50.0% of sample variance. Hence, to identify the lipid species accounting for the separation, each lipid class was analyzed individually by PCA (Appendix A). Briefly, the molecular species of each lipid class separately were able to differentiate, to a greater or lesser extent, primary cells from cancer cells. However, only PC, PE plasmalogens, and PS molecular species were able to separate Colo 201 from the rest of the cell lines.

Consistent with data in human colon epithelium [8], the most abundant PC species in all cell lines was 34:1 (34.6–50.9%, lowest and highest value throughout the five cell lines analyzed, respectively), followed by 36:2 (13.9–27.3%), 34:2 (6.8–13.1%), and 36:1 (7.4–9.2%). Within this lipid class, we detected an increase in 34:1 (34.6 vs. 44.0%, primary vs. the average value in cancer cells), and a decrease in 36:3 (5.4 vs. 3.3%) and in 36:2 (21.8 vs. 11,9%), except for Colo 201 that increased up to 27.3% (Figure 3A, Appendix A).

In PE, 36:2 (17.9–34.4%) was the most abundant species, followed by 36:1 (9.9–25.5%), 34:1 (13.0–15.9%), and 38:4 (4.7–14.3%). The increase in 40:7 and 40:6 (0.3 and 0.5% in primary vs. 2.9 and 4.7% in tumor cells, respectively) and the decrease in 38:3 (10.6 vs. 4.45%) were the most consistent changes throughout all cell lines (Figure 3B and Appendix A). In in situ cancer cell lines, 36:2, 38:3, and 38:4 levels were equally affected, with 36:2 (34.4 vs. 19.1% primary vs. mean value of all in situ cancer cells) and 38:3 (10.6 vs. 15.3%) decreasing, and 38:4 (8.9 vs. 13.1%) increasing. Considering that it is highly plausible that 18:0/18:2, 18:0/20:3, and 18:0/20:4 are the assignations for 36:2, 38:3, and 38:4, respectively, these changes would be highly consistent with the metabolic relationship existing within the fatty acids found at the sn-2 position. Conversely, in Colo 201, 38:4 levels were the lowest (4.7%), 36:2 levels were similar to the primary cells (32.1%), while 38:3 were similar to the in situ cancer cells (3.0%). Finally, 36:1 was increased in Colo 201 cells (1.8-fold) compared to all cell lines. In PE plasmalogens, the most abundant species was 16:0/20:4 (12.9–33.8%), followed by 18:0/20:4 (13.3–21.7%), 16:0/22:6 (0.7–26.9%), and 18:0/18:1 (5.1–18.1%) (Figure 3C and Appendix A). 

The most robust observation was the decrease in oleic acid (18:1)-containing species, either at the sn-1 or sn-2 position (16:0/18:1, 18:1/18:1, 18:0/18:1, 18:1/20:4), which coincided with changes reported in human colon tissue [15]. Consistent with previous reports, all cancer cells showed a significant increase in most DHA-containing species (16:0/22:6 and 18:0/22:6) [32].

The most abundant PI molecular species were 38:3 (13.2–47.9%), 38:4 (15.9–30.7%), and 36:1 (1.5–23.6%) (Figure 3D and Appendix A). The most noticeable changes occurred in MUFA-containing species (34:1 and 36:1). Whereas in primary cells, 34:1 and 36:1 accounted for approximately 1.0% total PI, in cancer cells the average values were 11.3% (5.1–15.6%) and 16.7% (9.6–23.6%), respectively. Interestingly, Colo 201 cells differed in their content in 34:1, 36:2, and 36:1 compared to the rest of cancer cells. In PS, the most abundant molecular species were 36:1 (37.8–60.5%), 36:2 (7.8–16.4%), and 34:1 (6.9–16.7%) (Figure 3E and Appendix A). The largest and most consistent change was the solid increase in 40:5 and 40:6 in all cancer cells compared to primary cells.

Regarding sphingolipids, in particular SM, Cer, and HexCer, the most abundant species were d18:1/24:1 (9.5–36.5% in SM, 10.9–26.7% in Cer, and 37.8–4.3% in HexCer) and d18:1/16:0 (36.7–58.2% in SM, 30.3–56.2% in Cer, and 10.1–66.1% in HexCer) (Figure 3F–H, Appendix A). Interestingly, the most profound changes occurred in these species, together with a significant increase in SMd16:1/18:1 in all cancer cells (2.7-fold). Thus, in cancer cells, d18:1/16:0 species increased (1.5-fold in SM, 1.2-fold in Cer 5.2-fold and in HexCer) while d18:1/24:1 species decreased (3.3-fold in SM, 1.5-fold in Cer, and 3.9-fold in HexCer) compared to primary cells, although these changes were not significant in all cancer cells. 

Next, we evaluated how a particular molecular species, expressed as the number of C-atoms: number of double bonds (38:1, 38:4, 40:5, etc.), were distributed within the main membrane phospholipid classes (PC, PE, PE plasmalogens, PS, and PI) (Figure 4, Appendix A). By doing so, we were able to evaluate into which phospholipid class cells channeled and stored a particular combination of fatty acids. At this point, we need to acknowledge that, except for PE plasmalogens, our lipidomic data did not provide the identity of each of the fatty acid moieties. Therefore, we had to assume certain assignations. To do this, we took into account the combination of fatty acids identified in PE plasmalogens, as well as data found in the literature. Nevertheless, alternative combinations of fatty acids contributing to the final percentage cannot be ruled out.

Interestingly, this way to express the results facilitated the identification of new cancer-associated alterations that, in this study, were common for all cancer cell lines. Thus, we were able to group the molecular species in four general categories according to the changes observed. The first group of species included 38:4, 40:6 36:4, and 38:6 species (Figure 4A, Appendix A) which, in cancer cells, were all drastically shifted from PE and PC to PE plasmalogens, and to PI in the case of 38:4. Importantly, this observation was independent of the cancer cell line and of the distribution in primary cells (e.g. 36:4 is different compared to 38:4), as in all cases the accumulation in PE plasmalogens accounted for up to 70% of the total. Indeed, there was a net increase in sn-1 saturated plasmalogen content in cancer cells (66% in primary vs. 93% in situ, 84% in Colo 201 cells, Appendix A). In the second group, which included 38:5 and 40:7 species, the distribution into PE plasmalogens was barely affected, with Colo 201 cells as the only exception (Figure 4B). This drastic difference between the first and the second group was rather unexpected as these species only differ in the presence or absence of a saturated fatty acid at the sn-1 position. The third group included 34:1, 36:2, 32:0, 34:2, and 36:3 species (Figure 4C, Appendix A), which clearly shifted to PC. Within this group, the changes in 36:2 and 36:3 species were significant, as the PC: PE distribution in primary cells shifted from 50:40 and 60:30 to 70:10 in cancer cells. Finally, the fourth group included 38:3, 40:5, 36:1, and 40:4 species (Figure 4D, Appendix A), which shared a large decrease in PE in all cancer cell lines. 

### 2.2. Levels of Plasmalogen are Coordinated with Enzyme Changes

A thorough analysis of the lipidome of these five colon cell lines offered two consistent results: a profound impact on the levels of ethanolamine glycerophospholipid subtypes (decrease in PE and increase in PE plasmalogens) and a drastic shift of AA- and DHA-containing diacyl species to PE plasmalogens. Unexpectedly, this shift mainly affected plasmalogens with saturated fatty acids at the sn-1 position. Since the latter is established during the early steps of ether lipid synthesis (Appendix A), we explored the differences between the primary and cancer cell lines in the expression of enzymes involved in the synthesis of fatty alcohols and their insertion at the sn-1 position of the glycerol backbone, namely: fatty acyl-CoA reductase (FAR, isoforms 1 and 2), glycerone-phosphate O-acyltransferase (GNPAT), and alkylglycerone-phosphate synthase (AGPS) (Figure 5, Appendix A).

Western blot analysis showed that FAR1, FAR2, and AGPS protein expression was consistently enhanced in all cancer cell lines compared to primary cells, although these differences were statistically different in all tumor cell lines simultaneously only for FAR1 (Appendix A). The largest impact was observed in AGPS expression, showing a 14- to 70-fold increase depending on the cancer cell line. Interestingly, AGPS was overexpressed in Colo 201 not only compared to primary cells but also, and importantly, compared to all the in situ cell lines, that is, SW480, LS174t, and HT29 cancer cells (1.5- to 5-fold higher). Interestingly, FAR1 and FAR2 were overexpressed in all cancer cells compared to primary cells, at rather similar levels for each cancer cell type (8- to 10-fold in Colo 201; 12-fold in HT29; 5-fold in LS174 and 8- and 12-fold in SW480). Finally, only SW480 cells showed an overexpression of GNPAT at the protein level (Figure 5A–D). Next, we evaluated the expression of these enzymes at the mRNA level by quantitative PCR. FAR1 mRNA levels were uniformly overexpressed in all cancer cell lines by 5.6-fold (average value). AGPS mRNA levels were also overexpressed in all cell lines, although values varied considerably: with a 4-fold increase in LS174t while the rest of cancer cells showed a 1.4-fold change. GNPAT mRNA levels were overexpressed by 3.6-fold in LS174t, while they remained constant in the rest of the cell lines (0.9-fold in Colo 201, 1.2-fold in HT29, and 0.9-fold in SW480). Finally, the FAR2 mRNA expression pattern was unexpectedly different between cells: overexpressed 7.0- and 2.1-fold in LS174t and HT29 cells, respectively, but downregulated by 0.3-fold in Colo 201 and SW480 cells, respectively (Figure 5E–H). Altogether these results were consistent with data on plasmalogen levels, indicating that the plasmalogen metabolism is profoundly affected at both the protein and the mRNA levels in all cancer lines (with the latter being highly dependent on the cell line). 

### 2.3. Extracellular Vesicle Lipids as Biomarkers of Malignization in a Cell Culture Model

EV lipid analysis was sensitive enough to separate them according to their cell origin. Thus, PCA of membrane lipid classes separated primary and cancer cell-derived EVs, although, unlike for cells, it was not able to discriminate between the in situ and Colo 201 cancer cells (Figure 6A). 

In any case, the most influencing lipid classes in the PCA were PC, SM, PE plasmalogens, and PS (Figure 6B). Thus, PCA results indicated that primary cells were segregated from the rest due to high PE plasmalogen levels and low PC levels (Figure 6B,C). Taken globally, in situ cell lines showed a profile with a high content in PC combined with low PE plasmalogen levels. Finally, the EVs of the most malignant cells were characterized by a low PC level compared to in situ cancer cells.

Consistent with the literature, the most abundant membrane lipids in EVs were PC (29.8–61.6%) and SM (28.4–35.4%) (Figure 6C, Appendix A) [26,27,33]. The most relevant differences between primary and cancer cells were the increase in PC (29.8 vs. 57.2% primary cells vs. tumor cells), and decrease in Cer (3.4 vs. 0.89%), PE (10.3 vs. 1.7%), and PI (4.1 vs. 0.7%) levels. Despite not being significant, it is worth stressing that EVs isolated from in situ cancer cells differed from those of Colo 201 in PS (1.5 vs. 10.3%) and PE plasmalogen (1.0 vs. 7.8% in situ cancer cells vs. Colo 201 cells) content. In terms of membrane lipid classes, EVs derived from in situ cancer cells were less diverse than those from the primary and Colo 201 cells. This homogeneity, together with the high PC and SM levels, would lead to more rigid membranes of the in situ derived EVs.

Next, we investigated the differences in cell and EV membrane lipidome by comparing their relative levels (Figure 6D). In this context, the most solid change was the rise in SM levels in EVs (4.1-fold enriched in EVs), which is in line with the literature [34]. The opposite was observed for PE, PE plasmalogens, and PS as all cell derived EVs were impoverished in these membrane lipids, whereas their levels between EVs and cells were maintained in the primary and Colo 201 cells. PI levels were rather similar in primary EVs and cells, while EVs derived from cancer cells contained lower levels of this phospholipid compared to their origin cell. Finally, the ceramide group was enriched in primary EVs (3.0-fold) but was barely affected in cancer cells.

The analysis of EVs lipid molecular species revealed profound changes in acyl chain composition in cancer cell lines compared to primary cells (Figure 7). The PCA using PC, PE, PI, and PS molecular species separately was able to separate between primary and cancer cells (Appendix A). Globally, the separation between primary and cancer cells was mainly due to their content in MUFA and DUFA species. The species accounting for this segregation were: high levels in PC32:0, 34:2, and 36:2; low levels in PE34:1 and 32:0; high levels in PI38:3 and low levels in PI36:1, and high levels in PS36:2 and 34:1, and low levels in PS36:1 content. Although the PCA indicated that PC34:1, PE36:2, and PI38:3 were increased in Colo 201 cells compared to in situ cancer cells, these differences were not statistical. The most abundant PC species in EVs were 34:1 (28.3–34.7%) followed by 36:2 (7.8–19.9%) and 36:1 (11.8–17.5%) (Figure 7A, Appendix A). Globally, the changes observed at the level of PC molecular species occurred in the same line and degree in both the in situ and metastatic cells, with PC34:1 as the only exception. Interestingly, 36:1 (a MUFA-containing PC) increased in cancer cells compared to primary cells (11.0 vs. 16.6% primary vs. mean cancer cell content). Conversely, primary cells were enriched in di-unsaturated species, 34:2 (12.0 vs. 2.5%) and 36:2 (19.9 vs. 8.8%), compared to all cancer cells.

In PE, the most abundant species were 34:1 (10.5–37.1%), 36:2 (24.0–42.4%), and 36:1 (15.7–23.7%) species (Figure 7B, Appendix A). The most relevant differences were the increase in 34:1 (10.5 vs. 30.3% primary vs. mean of all tumor cells) and 32:0 (0.4 vs. 10.0%), and the decrease in 38:3 (9.4 vs. 0.85%) in cancer cell derived EVs. In this study, fewer PE plasmalogen species were detected in EVs and their levels did not change statistically (Figure 7C, Appendix A). In PI, the most abundant species were 38:4 (29.1–36.7%) and 38:3 (14.9–43.5%) (Figure 7D, Appendix A). However, no significant differences were detected in PI species between the EVs derived from the cell lines studied. Finally, the most abundant PS species was by far 36:1 (52.6–66.1%) followed by 36:2 (6.5–16.3%) (Figure 7E, Appendix A). The most consistent changes throughout the cancer cell lines were the decrease in 38:3 (8.5 vs. 4% primary vs. mean of all tumor cells) and 40:4 (4.5 vs. 0.3%) content. Finally, regarding sphingolipids, d18:1/16:0 and d18:1/24:1 (36.8–45.4% and 17.4–23.2%, respectively) in SM, and d18:1/16:0 and d18:1/24:0 (19.9–30.6% and 16.5–34.3%, respectively) in Cer, were the most abundant SM species (Figure 7F,G, Appendix A). Taking into account these constant SM levels, it was rather remarkable to discover the increases in species of the likes of d18:1/22:1 and d18:1/22:0 in all “in situ” cancer cells and in d16:1/18:1, which increased similarly in all cancer cells. As in membrane classes, we compared how a particular molecular species was distributed between cells and EVs within each cell line (Appendix A). The most interesting result was the differential segregation of AA-containing species depending on whether this fatty acid was esterified in PC or PE. Thus, PC38:4 and 38:5 were preferentially channeled into EVs, while PE38:4 and 38:5 were directed into cells. Importantly, this observation was specific for cancer cells. 

## 3. Discussion

Cancer cell lines have been used in biomedical research worldwide for almost seven decades. Since the development of the HeLa cell culture in 1951, the use of immortalized cell lines implied a paradigm shift in the study of cancer [35]. Importantly, cancer cell lines helped to reveal the molecular mechanisms underlying multiple diseases, including, but not exclusively, cancer. Despite the controversy surrounding their use [36,37], cancer cell lines are still a valuable “in vitro” model system, widely used in cancer research, identification of biomarkers, and drug discovery [38,39,40]. Lipid research has been no exception in the use of this model, as could not have been otherwise. Taking into account that solid increase in the interest in lipidomics [41] and the solid results demonstrating the specificity of the lipid fingerprint, the use of commercial cell lines to investigate the role a particular lipid species is expected to rise. Herein, we demonstrate that the lipid analysis of five human colon cell lines rendered a lipid fingerprint that although, at first sight, may appear to be cell line-dependent, its comprehensive examination demonstrates that cancer and secreted EV lipidomes share common features that make it possible to distinguish not only between primary and cancer cells but also between cancer cell origin (primary tumor vs. metastatic site, Figure 1, Appendix A). Importantly, these changes were also specific enough to segregate cells according to their malignancy, firmly laying the groundwork for using both cell and EV lipid fingerprint as a CRC biomarker. Nevertheless, it is important to take into account the limitations of using cell cultures when interpreting lipidomic results [42], particularly those closely related to lipid metabolism, since it is highly sensitive media composition or culture conditions such as hypoxia/normoxia [43], cell density or cell growth type (2D vs. 3D cell cultures) [44].

Thus, the lipidomic data demonstrated that cancer and primary cell lipidome differ significantly at the level of both membrane lipid class and molecular species. Different lipid groups account for the separation between primary and cancer cells, with ethanolamine glycerophospholipids and sphingomyelin as the most critical (Figure 1, Appendix A). Hence, the most consistent changes throughout the cancer cell lines were the decrease in SM and PE and the increase in PE plasmalogens compared to primary epithelial cells, which fully agrees with the literature [42,43,44,45,46,47]. However, the causes of these alterations and the role of these lipids in the tumorigenic process remain unknown. 

SM appears to play a pivotal role in regulating cell adhesion [48], as well as cleavage furrow formation during cytokinesis [49], both critical processes in cell division and tumor development. Interestingly, our lipidomic results showed a common change throughout the sphingolipid family at the molecular species level. Thus, sphingolipids in primary cells were highly enriched in d18:1/24:1 species, while in cancer cells, they were enriched in d18:1/16:0. Although these are rather common sphingolipid species, there is still no clear evidence regarding their specific role, which could depend on cell type. Thus, SMd18:1/16:0 was identified as a biomarker for hypoxic tumor regions in a breast cancer xenograft model [50], whereas in liver cancer [51], this species enabled common necrosis (typical of tumor progression) to be distinguished from infarct-like necrosis (a response to treatment). Unlike in phospholipids, fatty acid turnover is not a mechanism used by mammalian cells to modify sphingolipid molecular species, so the alterations in lipid metabolism must be happening at the species de novo synthesis or degradation level in cancer cells.

Further, the comprehensive analysis of the lipid fingerprint left us with two solid results affecting ethanolamine glycerophospholipids on two levels. First, ethanolamine glycerol-phospholipids were affected at the level of subclasses. Thus, we established a profound impact on PE and PE plasmalogen levels, decreasing and increasing, respectively. The first study reporting enhanced ether lipid levels in cancer dates back to the late 1960s [45]. Since then, many studies, including ours, have consistently shown higher levels of ether lipids in cancer cell lines [45], in xenograft models [47] or human tumor samples [45]. According to our results, the increase in PE plasmalogens could be explained by overexpression of the key plasmalogen synthesis enzymes, FAR1, FAR2, and AGPS, in all cancer cell lines. Importantly, despite the solid evidence linking plasmalogens and cancer, little is known about the impact on ether lipid synthetic enzymes of the tumorigenic process. The initial reaction in ether lipid synthesis requires the formation of a complex between two AGPS and one GNPAT molecule to acylate dihydroxyacetone phosphate at the sn-1 position [52] (Appendix A). Recently, AGPS overexpression was associated with cancer cell aggressiveness and invasiveness in primary breast tumors and cancer cell lines (breast (231MFP), melanoma (C8161), and prostate (PC3) cancer cells) [45]. Consistently, we showed that not only was AGPS overexpressed in all cancer cell lines but also between cell lines isolated from primary tumors or metastatic sites (Figure 5C). Taking into account the fact that the levels of ether lipids are positively associated with the tumor metastatic capacity of a cell line [45,46], our results consolidate AGPS expression as a good biomarker to assess the metastatic capacity of a cancer cell. Conversely, GNPAT expression was only overexpressed in LS174T, while in the rest of cells it was either unaffected or slightly downregulated. A similar observation, a reduction in GNPAT mRNA levels, was reported in a study on murine microglia exposed to inflammatory stimuli [53]. Therefore, it cannot be ruled out that the tumor inflammatory component could account for the GNPAT mRNA levels observed in this study. To the best of our knowledge, no data comparing GNPAT, FAR1, and FAR2 expression in tumor and non-tumor cell lines have been published to date.

The second most striking impact on ethanolamine glycerophospholipids was at the level of molecular species. Herein, we report for the first time a profound shift of PE and PC diacyl species into PE plasmalogens, which proved to be highly selective for those presenting saturated fatty acids and AA (36:4, 38:4) or DHA (38:6, 40:6) at the sn-1 and sn-2 position, respectively. Despite the fact that plasmalogens are the often-unperceived member of phospholipid family, these results clearly indicate that they are more tightly regulated than initially suspected. Fatty acyl-CoA reductases (FAR1 and FAR2) are the enzymes catalyzing the reduction of a fatty acyl-CoA to the fatty alcohol that is finally linked at the sn-1 position of the plasmalogens [54]. FAR1 is the isoform showing the broadest distribution, whereas FAR2 expression is largely restricted to the meibomian glands, skin, brain, and small intestine (no data have been reported for colon) [55]. It has been proposed that differences between enzyme expression, activity, and substrate preference could be related to a cell strategy to differentially channel fatty alcohols to ether lipid or wax ester synthesis [56]. Importantly, FAR1 and FAR2 also differ in their activity and substrate preference. FAR1, which seems to be more active, accounts for C16:0, C18:0, and C18:1 fatty alcohol synthesis, while FAR2 prefers C16:0 and C18:0 saturated fatty acids [57]. Furthermore, FAR1 stability (but not FAR2) is regulated post-translationally by a mechanism sensitive to plasmalogen levels, wherein the adequate localization of plasmalogens in the inner leaflet appears to be fundamental [57,58]. 

Using duramycin, a probe that binds both diacyl- or plasmalogen type [59], it was demonstrated that apoptosis or exposure to noxious cues induces translocation of ethanolamine glycerophospholipids [60,61,62]. Hence, it cannot be ruled out that the translocated PE could contain a fraction of PE plasmalogens. If so, this could trigger a “low plasmalogen level” signal at the inner leaflet [63] and leading to the overexpression of FAR1 levels (Figure 5). Conversely, in the context of exacerbated plasmalogen synthesis, FAR2 fatty alcohols might also be used in plasmalogen synthesis. This could be particularly important in the context of intestinal cancer were FAR2 mRNA levels have been shown to be high compared to other tissues [55]. It is worth stressing that despite the individual differences in the lipidome of cancer cells, when compared to primary cells, the lipid and protein changes reported herein were strikingly homogeneous in the four cancer lines, pointing to what could be a common feature of the tumorigenic process. Regarding the specificity at the sn-2 position, it is well known that AA is preferentially incorporated and accumulated at this position in PI and PE plasmalogens, although the biological implication of this specificity remains unestablished [64,65]. In this context, we have recently demonstrated that PE plasmalogens and PI molecular species containing AA are strictly regulated during colonocyte differentiation [8,15]. Thus, AA-PE-plasmalogen/PI levels are highest at the base of the crypt, where continuous stem cell division occurs, and they decrease concomitant to the advance in the differentiation process. Importantly, these changes were accompanied by a gradient in the expression of enzymes involved in AA metabolism. Further, higher levels of AA-containing phospholipids have been consistently shown in culture cells [32,45,46,66] as well as in colon adenomatous polyps and carcinoma [8,15]. Altogether, we hypothesize that rapidly dividing cells, whether they be stem cells or cancer cells, need to accumulate and preserve esterified AA levels, particularly in PI and PE plasmalogens, to sustain their division rate, which would account for the robust shift observed in this study. Figure 8 summarizes our current understanding of how the observed cancer-induced changes in plasmalogens may participate in exacerbated cell division due to their involvement in the recruitment and activation of Akt, a key regulator of cell proliferation [65,67].

There is no doubt that detecting cancer at very early stages is a factor critical to patient survival. Therefore, the possibility of detecting differential EV profiles in terms of composition has led many researchers to characterize their content thoroughly. However, despite the general interest in this field, few papers describe the complete lipidome of EVs [26,33]. In this study, the results showed that the EV lipidome is dependent on cell origin, and therefore has the potential of becoming a good biomarker. However, the comprehensive description of the EV lipidome of five different colon cells lines enabled us to conclude that it was possible to segregate between EVs shed by cancer cells from those shed by non-cancer cells. These changes could help in future studies to understand the role and regulation of EVs, but more importantly, could set the basis to use the EV lipidome as a non-invasive cancer biomarker tool. The results in terms of phospholipid composition are in agreement with previous studies showing a relevant SM enrichment in EVs compared to cell of origin [34]. While several studies describe the lipidome of EVs obtained from rather different sources [18,19,20,21,22,23,24], only a few studies explored how EV lipid molecular species could be used as biomarkers [25,26,27]. Interestingly, we were able to segregate EVs according to their origin by using lipid PC, PE, PI, and PS molecular species confirming the great potential the EV lipidome has in generating biomarkers for cancer diagnosis.

## 4. Materials and Methods

### 4.1. Cell Lines and Culture Protocol

CRC cell lines, Colo-201, HT29, LS174T, and SW480 were purchased from the European Collection of Authenticated Cell Cultures (ECACC, Salisbury, UK). To avoid contamination by serum-EVs, all cell lines were cultured with EVs-free serum that was prepared by centrifuging the commercial serum at 120,000× *g* overnight. The primary colon epithelial cell line was purchased from Innoprot (Ref. P10768, Derio, Spain). Cells were grown for 72 h in the medium recommended by the manufacturer and completed, except for primary cells, with 10% v/v exosome-free FBS (Labclinics, Barcelona, Spain, Ref. S181B-500), 1% v/v non-essential amino acids (Labclinics, Ref. NEAA-B), and 2 mM glutamine (Labclinics, Ref. P1012-500GR). Cells were plated and grown as follows: primary cell line, 30,000 cells/cm^2^ in Colonic Epithelial Cell Medium (Innoprot, Bizkaia, Spain, Ref. P60165) for 48h; SW480, 20,000 cells/cm^2^ in DMEM for 48 h (Labclinics, Ref. L0106-500); HT29, 30,000 cells/cm^2^ in MEM for 48 h (Labclinics, Ref. MEMA-RXA); LS174t, 30,000 cells/cm^2^ in MEM for 48 h (Labclinics, Ref. MEMA-RXA); and Colo 201, 20,000 cells/cm^2^ in RPMI for 72 h (Labclinics, Ref. L0490-500). For EV collection, the culture medium was collected 48 h after being plated (72 h for the Colo 201) and kept at −80 °C until processing. Protein levels were measured using a protein assay based on the Bradford dye-binding method, according to the manufacturer’s instructions (Bio-Rad Laboratories, Barcelona, Spain).

### 4.2. Extracellular Vesicle Isolation

EVs were isolated using a differential centrifugation protocol adapted from Crescitelli et al. [29]. The first centrifugation was at 300× *g* for 10 min to precipitate the floating cells and cell debris. Then, the supernatant was centrifuged at 2000× *g* for 20 min. to precipitate the apoptotic bodies. Next, the supernatant was filtered by gravity through 0.8 µm filters to remove undesired particles > 800 nm. The supernatant was filtered with gentle pressure to isolate the smallest vesicles. Finally, the supernatant was centrifuged at 120,000× *g* for 70 min. to obtain the exosome-enriched fraction. All centrifugation steps were performed at 4 °C.

### 4.3. Lipid Extraction and LC-MS

#### 4.3.1. Lipid Standard Solutions

Internal standard solutions were prepared as described previously [73]. For measurements of PE P-, GM1, and GD1, an external standard solution containing 50 pmol each of PE (28:0), PE (P-18:0/18:1), GM3 (d18:1/12:0), and GD1a (from bovine brain) was prepared. Pretreatment and measurement of the external standard solution were performed simultaneously with the samples, and derived (peak area of PE (P-18:0/18:1)/peak area of PE (28:0)) values were used as the corrective coefficients for the quantitation of PE plasmalogen.

#### 4.3.2. Lipid Extractions

Lipid extractions were performed as described previously [73], but with several modifications. Briefly, cell pellets were sonicated for 10 s with 0.1 mL methanol/butanol (1:1) to inactivate the associated enzymes using an ultrasonic bath. After the addition of 0.05 mL standard lipid mixture, 0.05 mL of 0.5 M phosphate buffer (pH 6.0), and 0.2 mL of water, the samples were shaken with 0.7 mL of butanol and sonicated for 3 min. in an ultrasonic bath. After centrifugation, the upper layer was collected. The original suspension was re-extracted by the addition of 0.35 mL each of ethyl acetate and hexane, followed by centrifugation. The resulting extract was combined with the first butanol extract. After the addition of 0.7 mL methanol, 10% (0.21 mL) of this solution was dried under reduced pressure at 40 °C, and dissolved in 20 μL of LC mobile phase B and 30 μL of mobile phase A. This sample was used to analyze Cer, SM, monohexosylceramide (HexCer), PE, and PC levels. The remaining 90% (1.8 mL) of the extract was fractionated on a DEAE-cellulose column (500 μL bed volume packed in a 1 mL polypropylene pipette tip), previously activated by acetic acid. After washing with 2 mL of methanol, the column-bound lipids were eluted with 1 ml methanol/28% aqueous ammonia/formic acid (1000:33:22). The organic solvent was evaporated from the eluate under reduced pressure at 50 °C, after which dried materials were dissolved with 50 μL of mobile phase A. The resulting sample was used for the analyses of acidic lipids (i.e., GM3, PS, PG, PI, and phosphatidic acid (PA)).

### 4.4. MS Analysis

Lipids were measured using LC-MS/MS as described previously [73], except that the collision energy was set to 30 V for Cer, SM, HexCer, LacCer, PE, and PC. Mass transitions were additionally set to 702.5/364.2, 724.5/364.2, 728.6/390.2, 730.6/392.2, 748.5/364.2, 750.5/390.2, 752.6/392.2, 774.5/390.2, and 776.6/392.2 for PE plasmalogens in the positive ion mode. Each molecular species was identified based on the MS/MS spectrum and LC retention times, and quantities present were calculated from the peak areas of the measured lipids, compared with those of the internal standards. Each level of measured lipids was normalized to the total protein content.

### 4.5. Protein Expression Determined by Western Blot

Samples were lysed using protein extraction buffer (10 mM Tris-HCl pH 7.4, 50 mM NaCl, 1 mM MgCl_2_, 2mM EDTA, 1% w/v SDS, Sigma-Aldrich, Madrid, Spain) with complete protease inhibitor tablets (Roche, Basel, Switzerland). Sample lysates were briefly sonicated (450 Digital Sonifier, Branson, Hampton, NH, USA) at 4 °C and protein concentration was measured using the DC Protein Assay Kit (Bio-Rad, Barcelona, Spain). Then, 10% loading buffer was added to the sample which was boiled for 8 min. and a total of 20 μg protein was loaded for analysis in an SDS-PAGE. After protein transfer, nitrocellulose membranes (protran ba85, GE Healthcare Life Science, Barcelona, Spain) were blocked with PBS (0.14 M NaCl, 2.7 mM KCl, 8 mM Na_2_HPO_4_, 2 mM KH_2_PO_4_) containing 5% dry non-fat milk for 1 h at room temperature. Membranes were then incubated at 4 °C overnight with primary antibodies against GNPAT (1:600, Atlas Antibodies, Stockholm, Sweden, cat# HPA060059), AGPS (1:600, Atlas Antibodies, cat# HPA030209), FAR1 (1:6000, Antibodies-online, cat# ABIN2174097), FAR2 (1:6000, Antibodies-online, cat# ABIN709091), and β-actin (1:10,000, LI-COR Biosciences Lincoln, NE, USA, cat# 926-42212). After incubation, membranes were washed with PBS containing 0.1% Tween 20, or PBS containing 0.1% Tween 20 and 5% BSA (for polyclonal antibodies) (Sigma-Aldrich). Then, membranes were incubated with goat anti-rabbit IRDye 800CW (1:5000, LI-COR, cat# 926-32211) or Alexa 685 donkey anti-mouse IgG (1:2500, Abcam, Cambridge, UK, cat# ab175774) secondary antibodies at room temperature for 1 h. Membranes were visualized using Odyssey CLx Imaging System (LI-COR Biosciences); Quantity one software (Bio-Rad, Hercules, CA, USA) was used to quantify the specific signals.

### 4.6. Statistical Analysis

Cell lines and derived EVs were compared in each case using one-way ANOVA followed by the Bonferroni multiple-comparison post-test. Values are expressed as mean ± SD values from 3-6 independent experiments. 

## 5. Conclusions

There is no doubt that tumorigenesis is a complex process which affects cell lipidome at different levels. Consistently, the lipid profile has proven to be specific enough to unambiguously characterize different cell states such as division, differentiation, malignization, and cell death [8,15,47,74,75] which makes the lipidome a powerful tool to identify biomarkers for disease. Furthermore, the identification of specific molecular species rigorously regulated during fundamental biological processes is leading to a scenario wherein the specific role of membrane lipids needs to be redefined and further investigated. Here, we demonstrate that even though it is important to be aware of the differences in individual lipidomes between cancer lines, even if they share tissue origin, commercial cell lines do show striking homogenous changes, pointing to what could be a common feature of the tumorigenic process. This conclusion is particularly important when developing new drugs targeting lipid metabolism in a clinical setting. Thus, cell cultures are still a valid model to study lipid metabolism, but caution should be taken due to its characteristics and its high dependence on cell culture conditions. Individual differences within tumor CRC cell lines are, in fact, in line with the scenario found in CRC patients, where it is well established that, because of compositional differences, CRC tumors at the same stage can differ in both prognosis and treatment response [3]. Consistently, there is a great effort to resolve this heterogeneity by defining a molecular sub-classification [3]. Taking into account the sharp rise in lipidomic articles in the last decade, the increase in the number of studies using commercial cell lines aiming to identify the exact molecular mechanisms used by these molecular species is more than likely. Hence, when interpreting results, it is critical to take into account that any culture conditions (cell media composition, hypoxia [47,76], cell confluence, or 2D vs. 3D cultures [77]) may have an impact on the lipidome.

The increasing number of studies demonstrating the high versatility of lipid metabolism to describe pathological situations places lipidomics at a crucial moment. Even though the use of lipidomic analysis in a clinical setting is currently still discrete, some examples include the diagnosis of severe metabolic diseases in newborns by detecting aberrant levels of very long polyunsaturated fatty acids, or malnourishment by assessing n-3 fatty acids in plasma. This corroborates the importance of having a thorough understanding of lipid metabolism and how it is regulated at all levels (enzyme activity, gene expression, and epigenetically). The incorporation of MS-based methods into the research lab routine, together with an improvement in the study models (e.g. use of organoids), allows being optimist about the use of lipidomics in daily clinical practice soon. Proof of that is the great effort being made by international consortia, such as Lipid Maps, to develop and establish harmonized protocols for lipid analysis and data mining.

## Figures and Tables

**Figure 1 cancers-12-01293-f001:**
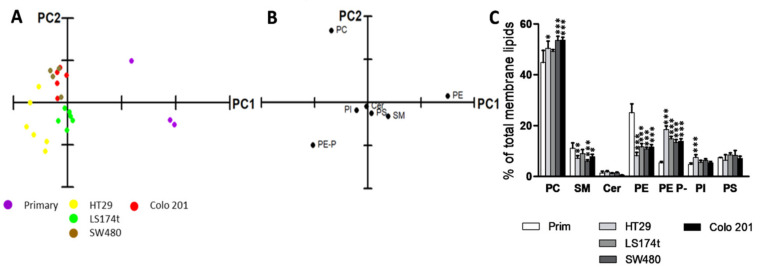
Analysis of the main membrane lipid classes of the commercial cell lines analyzed. (**A**) PCA using membrane lipid levels expressed as % of total membrane lipids. Explained Variance = 83.4%; (**B**) Loading plot after PCA of the main membrane lipid classes. For clarity, only the most influential species are indicated in each variable PCA analysis; (**C**) Membrane lipid composition. Values are expressed as % of total membrane lipids (mean ± SD), *n* = 3–6. Statistical significance was assessed using one-way ANOVA followed by Bonferroni post-test. For clarity, only statistical differences between primary and cancer cells are represented. The asterisk (*) indicates a significant difference between cancer cell lines and the primary cell line. * *p* < 0.05; ** *p* < 0.01; *** *p* < 0.001. Detailed results showing all comparisons are included in Appendix A.

**Figure 2 cancers-12-01293-f002:**
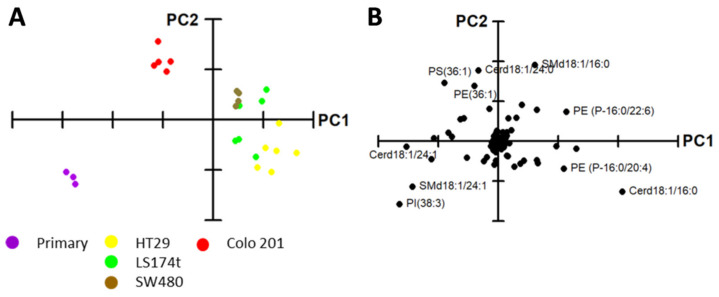
Cell lipidome segregates cell lines according to their malignancy. (**A**) PCA using the levels of all lipid species expressed as % of total lipid class. Explained variability 54.6%; (**B**) Loading plot after PCA of the main membrane lipid classes. For clarity, only the most influential species are included.

**Figure 3 cancers-12-01293-f003:**
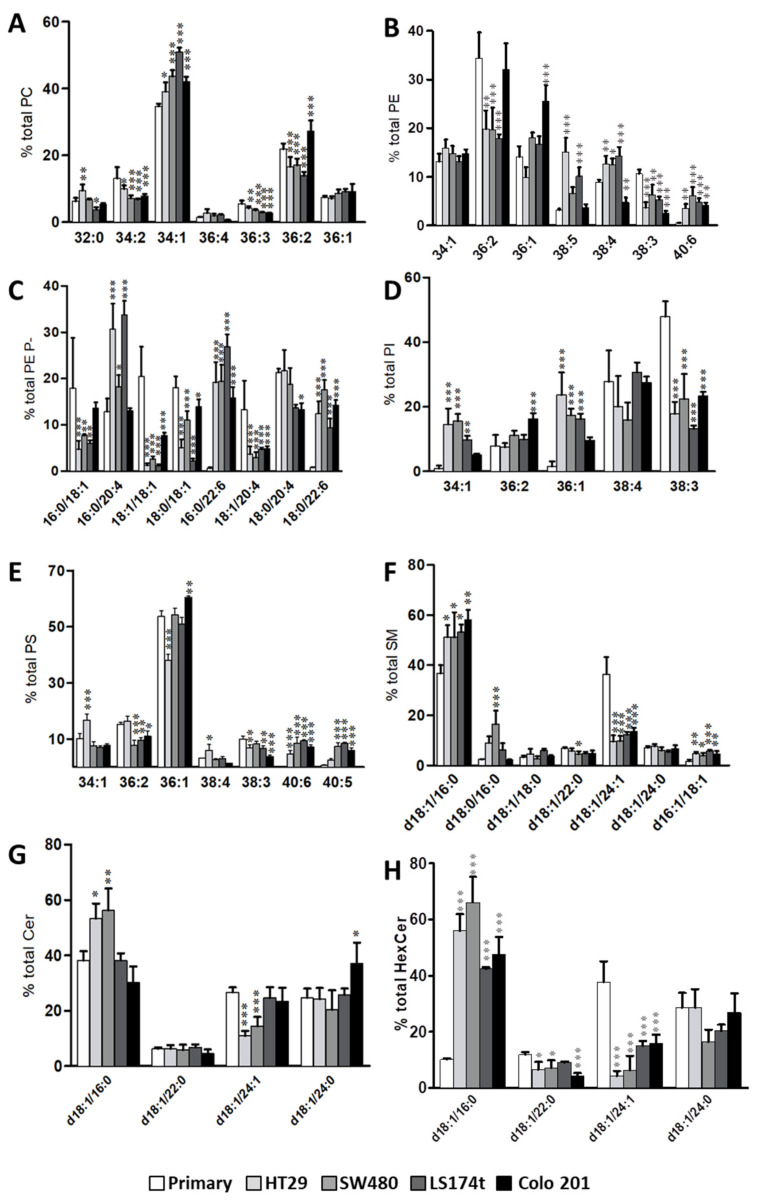
Membrane lipid fingerprint of primary, in situ, and metastatic cancer cell lines. Bar diagrams comparing changes in lipid composition of (**A**) PC, (**B**) PE, (**C**) PE plasmalogens, (**D**) PI, (**E**) PS, (**F**) SM, (**G**) Cer, and (**H**) HexCer at the molecular species level in primary, HT29, LS174t, SW480, and Colo 201 cell lines. Values are expressed as percentage of total fatty acid (mole %) and represent mean ± SD, *n* = 3–6. Statistical significance was assessed using one-way ANOVA followed by Bonferroni post-test. For clarity, only significance with respect to primary cells are expressed, * *p* < 0.05; ** *p* < 0.01; *** *p* < 0.001; and only species accounting for <5% of total membrane lipid class are included in the graph. Detailed results of all comparisons and all lipid species are included in Appendix A.

**Figure 4 cancers-12-01293-f004:**
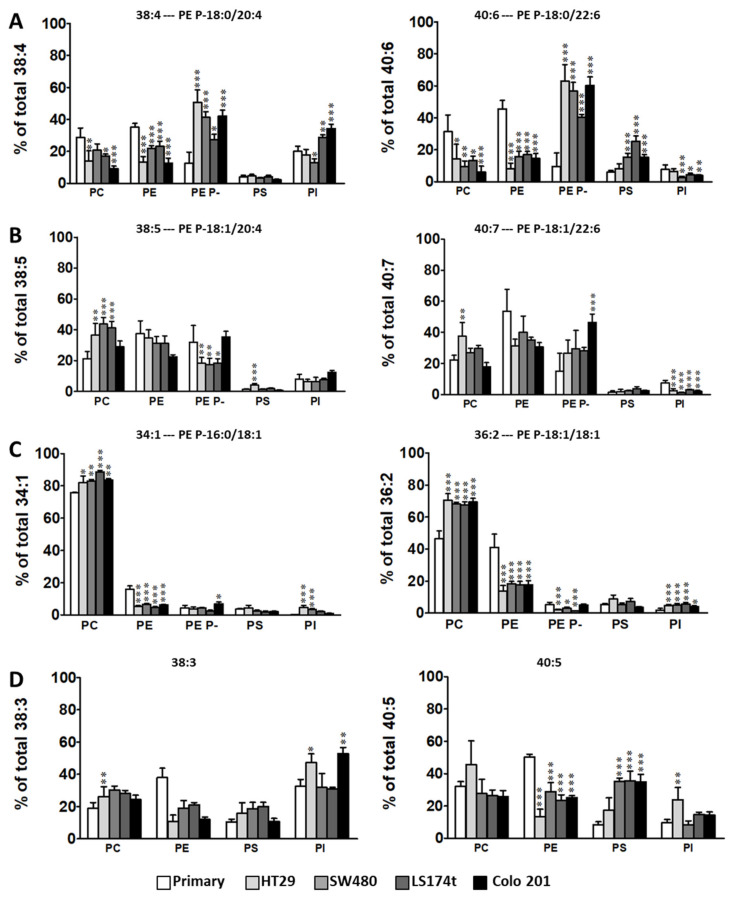
Specific shift of PC and PE molecular species to sn-1 saturated /sn-2 AA or DHA - containing PE plasmalogens in cancer cells. The distribution of the total amount of a particular fatty acid combination within each membrane phospholipid class was evaluated. (**A**) 38:4 and 40:6-containing phospholipids; (**B**) 38:5 and 40:7-containing phospholipids; (**C**) 34:1 and 36:2-containing phospholipids; (**D**) 38:3 and 40:5-containing phospholipids; Values are expressed as a percentage of the total amount of the selected fatty acid combination (mole %) and represent mean± SD, *n* = 3–6. Statistical significance was assessed using one-way ANOVA followed by Bonferroni post-test. Only significance with respect to primary cells are expressed. * *p* < 0.05; ** *p* < 0.01; *** *p* < 0.001. Detailed results of all comparisons are included in Appendix A. Minor species are included in Appendix A.

**Figure 5 cancers-12-01293-f005:**
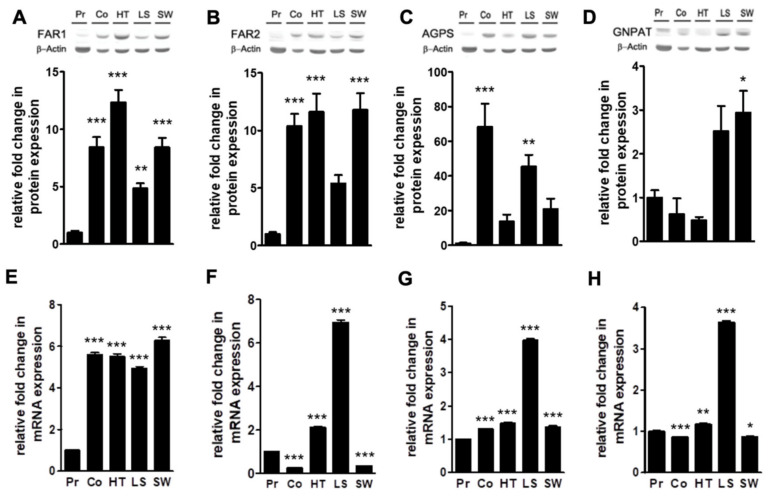
Protein and gene expression of ether lipid synthetic enzymes in primary and cancer colon cell lines. (**A**–**D**) Protein expression of ether lipid synthesis enzymes: FAR1 and FAR2 (fatty acyl-CoA reductases 1 and 2), AGPS (alkyl-glycerone-3-phosphate synthase), and GNPAT (glyceronephosphate O-acyltransferase) in primary (Pr) and cancer colon cell lines (Co, Colo-201; HT, HT29; LS, LS174T; SW, SW480). Values are expressed as a percentage of control and represent the mean ± SEM, *n* = 3–5; (**E**–**H**) Gene expression of ether lipid synthesis enzymes in primary (Pr) and cancer colon cell lines (Co, Colo-201; HT, HT29; LS, LS174T; SW, SW480). Values are expressed as a percentage of control and represent the mean ± SEM, *n* = 5. To assess statistical differences, one-way ANOVA and Bonferroni post-test were applied. For simplicity only significance with respect to primary cells are expressed. * *p* < 0.05; ** *p* < 0.01; *** *p* < 0.001. Detailed results of all comparisons are included in Appendix A. Original values and densitometry values are included in Appendix A and Appendix A, respectively.

**Figure 6 cancers-12-01293-f006:**
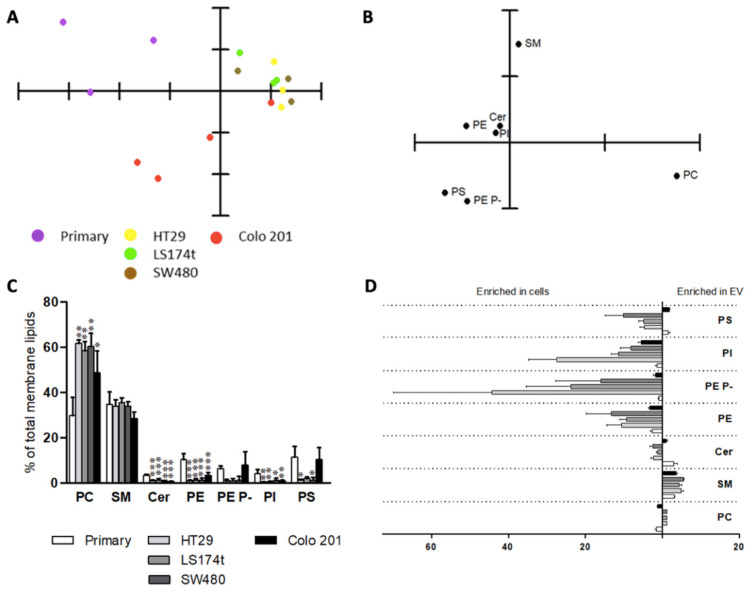
Membrane lipid composition of EVs isolated from colon commercial cell lines. (**A**) PCA of the EV lipid composition at the level of membrane lipids classes; (**B)** Loading plot after PCA of the major membrane lipid classes. For clarity, only the most influential species are indicated at each variables PCA analysis; (**C**) EV membrane lipid composition. Values are expressed as % of total membrane lipids (mean ± SD), *n* = 3–6. Statistical significance was assessed using one-way ANOVA followed by Bonferroni post-test. For clarity, only statistical differences between primary and cancer cells are represented. The asterisk (*) indicates a significant difference between cancer cell lines and the primary cell line. * *p* < 0.05; ** *p* < 0.01; *** *p* < 0.001. Detailed results showing all comparisons are included in Appendix A. (**D**) Membrane lipid class segregation between cells and cell-derived EVs. Enrichment of lipid classes in cells or exosomes calculated as mol% of lipids in these samples.

**Figure 7 cancers-12-01293-f007:**
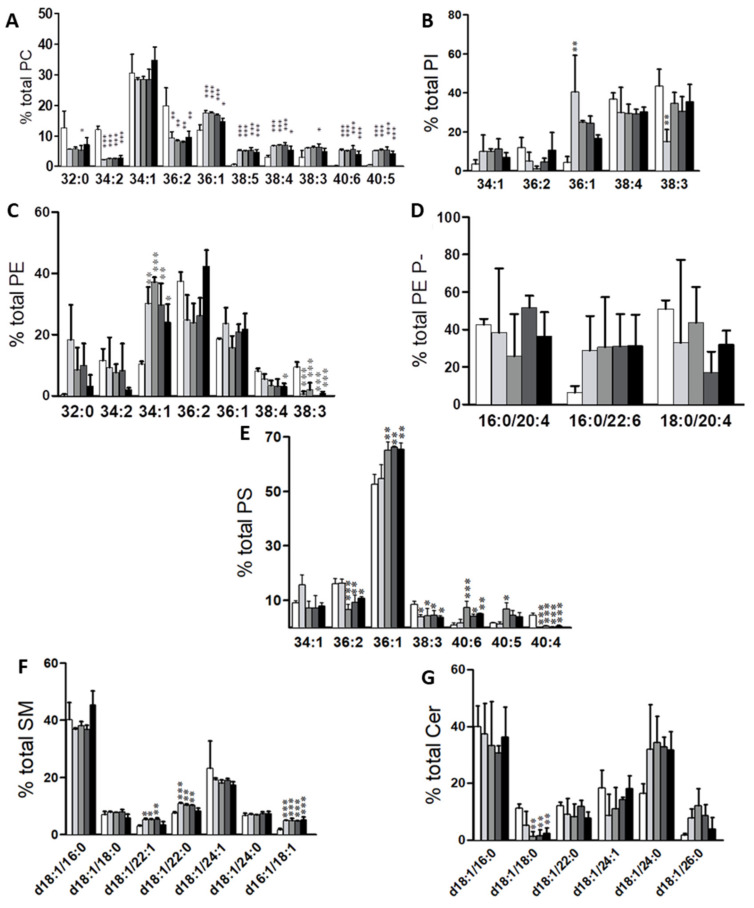
Molecular species composition of the main membrane lipids of EVs isolated from commercial colon cell lines. Bar diagrams comparing levels of (**A**) PC, (**B**) PI, (**C**) PE, (**D**) PE plasmalogens, (**E**) PS, (**F**) SM, and (**G**) Cer at the molecular species levels in primary, HT29, LS174t, SW480, and Colo 201 cells. Values are expressed as a percentage of total fatty acid (mole %) and represent the mean ± SD, *n* = 3–6. Statistical significance was assessed using one-way ANOVA comparing primary to cancer cells. For clarity, only species accounting for <5% of the total lipid class were included in the graphs. * *p* < 0.05; ** *p* < 0.01; *** *p* < 0.001. Detailed results of all comparisons are included in Appendix A.

**Figure 8 cancers-12-01293-f008:**
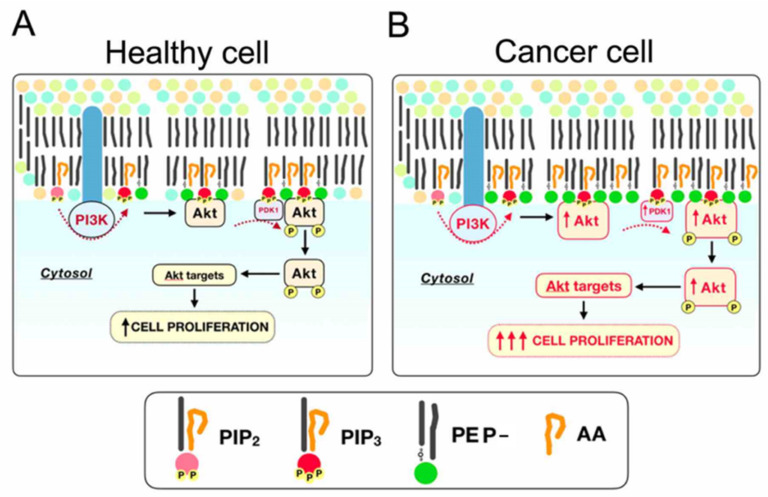
Model describing the impact of the most consistent lipid changes observed in cancer cells—increase in PE plasmalogen- and AA-containing phospholipids (in particular PI)—on the Akt signaling pathway, a canonical cell differentiation and proliferation pathway. (**A**) In healthy cells, phosphatidylinositol-3–kinase (PI3K) phosphorylates PIP2 to PIP3, which recruits Akt directly via a PH-pleckstrin domain. Despite the lack of direct evidence indicating the preference of PI3K enzymatic for AA-containing PIP2, this specificity was shown for PI4K [68,69]; in addition, both PIP2 and PIP3 are enriched in AA [70]. Altogether, it could be speculated that PI3K, may prefer AA-containing substrates. Thus, other papers show that plasmalogens are needed to maintain Akt linked to the membrane [67,71], which is crucial for its activation via phosphorylation by PDK1 and PDK2 (among others). Once phosphorylated, Akt shuttles back to the cytosol where it phosphorylates a myriad of targets, activating downstream pathways that culminate in cell proliferation. (**B**)—In cancer cells, including colorectal cancer cells, PI3K and Akt are overexpressed at the protein level [72]. Therefore, the presence of high levels of AA-containing phospholipid, and plasmalogen [8,15,45,46,47] in cancer cells would provide the substrate and necessary environment to sustain enhanced and uncontrolled cell division.

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
