# Peer review of "Common and Differential Traits of the Membrane Lipidome of Colon Cancer Cell Lines and Their Secreted Vesicles: Impact on Studies Using Cell Lines"

_cancers, 2020, doi:10.3390/cancers12051293_

Round 1
Reviewer 1 Report
This is an interesting study that aimed to characterize the membrane lipidome of five commercial colon cell lines and their extracellular vesicles. In general, the manuscript is well written; please check throughout the text for grammar and spelling errors. Methods section is clear and well described. Figures and tables are detailed and helpful for the reader. I would suggest to include a brief discussion on the importance for clinical studies to consider serial measurements of tumor lipids to prove target modulation (i.e. Jones DT et al, Mol. Cancer Ther. 2019), and on the impact of hypoxia on lipid metabolites (i.e. Valli A et al. Oncotarget 2015). Eventually, I would include further discussion in the conclusion paragraph on the future direction and possible application of the results.
Author Response
We appreciate all the comments of the reviewer very much. We have thoroughly reviewed the manuscript searching for spelling mistakes. They are all marked in red in the reviewed version.
Point 1:
I would suggest to include a brief discussion on the importance for clinical studies to consider serial measurements of tumor lipids to prove target modulation (i.e. Jones DT et al, Mol. Cancer Ther. 2019), and on the impact of hypoxia on lipid metabolites (i.e. Valli A et al. Oncotarget 2015).
Response 1:
We agree with the reviewer, hypoxia is a very important aspect to take into account when investigating lipid metabolism. It has been well studied in the context of brain ischemic injuries and of cancer, as tumor lesions have hypoxic areas. Hence, we have included the concept of hypoxia and growth type (2D vs. 3D cell cultures) as one of the aspects altering the lipidome (line 390 and line 613).
Point 2:
Eventually, I would include further discussion in the conclusion paragraph on the future direction and possible application of the results.
Response 2:
Finally, as recommended by both the reviewers, we have included a paragraph in the conclusions to explore future directions and applications of the results. See last paragraph in the Conclusion Section.
Reviewer 2 Report
General Comment:
The work by Bestard-Escalas et al. presents an in depth lipidomic analysis of colorectal cancer (CRC) cells and extracellular vesicles. The manuscript is very well organized and has very good readability despite the vast amount of work and data presented. The statistical analysis was detailed and the methods appropriate. This is an important contribution to establish lipid biomarkers for early diagnosis of CRC, and to evaluate its evolution and malignancy: It will therefore contribute to develop new valuable tools in the clinical contexto. Furthermore, it highlights the importance of the lipid profile and its tight regulation in tumorigenesis and other pathophysiological processes, stimulating further research on the regulation mechanisms and biological roles of individual lipid species.
Minor remarks:
There are no results for cholesterol, which is a major membrane lipid. Could the Authors comment on this? If there were no significant variations, this should be mentioned. The same applies to gangliosides, which were also analyzed, according to the Experimental Section. Also, related to this, the titles for Tables S1 and S2 in Supplementary Material could be misleading, (e.g., Membrane lipid composition).
Can the Authors detail slightly the reasons for choosing the extraction methods used?
In the Discussion, it would be interesting to read something about the expectations regarding the applicability of the results presented in clinical practice (e.g., possibilities of automation, cost-benefits, where does it stand among the other available tools – advantages/complementarities, etc.) and whether protein level and/or mRNA levels of the enzymes in PE-P metabolism could be more advantageous in comparison to the lipidomic analysis.
Sentences deserving revision/clarification:
L-61-64 “In this sense, our previous study analyzing the changes in the lipid signature along the colon…”
L.89 “Altogether, these results verify the capacity of lipid profiles, whether from cells or from EVs, for 89 perceiving physiological alterations, such as those occurring during tissue malignization, and 90 consequently, in providing potential lipid biomarkers.”
Maybe change to “These results allow to verify … and to perceive…”
L. 167 “within the fatty acids found at the sn-2 fatty acid”
At the sn-2 position?
L. 168 “while 38:3 decreased to similar to the in situ cancer cells”
L. 174 “Consistent with previous reports40”
L. 256 “Interestingly, FAR1 and FAR2 were overexpressed in all cancer cells compared to primary cells, interesting at rather similar levels for each cancer cell ty”
Please correct typos.
L. 285 “Just as with the cell lipidome, EV lipid analysis was sensitive enough to separate them according to their cell origin”
The sentence does not read as one in the beginning of a sub-section
L. 289 “Primary cells were separated from the rest due to high PE plasmalogen levels and low PC levels”
Please rephrase sentence for clarity
L. 312 “The latter together with the high PC and SM levels would imply that in situ-EV’s would lead to more rigid membranes.”
Please clarify. This comparison is made between EV and plasma membrane of Colo201 cells?
L. 322 “Next, we investigated the differences in membrane lipid composition cell and EV lipidome”
L. 360 “As in membrane classes, we compared line how a particular molecular species”
L. 398 “whereas in liver cancer64”
L. 442: “C18:0 saturated fatty acids69”
L. 445-447 “Considering that apoptosis or exposure to noxious cues induces translocation to the outer leaflet of both diacyl and ether ethanolamine glycerophospholipids [56–58], it is possible that in cancer cells, the unusual translocation of PE plasmalogens could wrongly convey “a low plasmalogen level” signal, leading to overexpressed FAR1 levels, as reported in this study (Figure 5). “
What do the authors mean with unusual translocation – there is evidence for exposure of PE-P in the outer leaflet in cancer cells? Also, it seems to be in contradiction with the mechanism proposed in Figure 8. Please clarify.
Author Response
Point 1:
There are no results for cholesterol, which is a major membrane lipid. Could the Authors comment on this? If there were no significant variations, this should be mentioned. The same applies to gangliosides, which were also analyzed, according to the Experimental Section. Also, related to this, the titles for Tables S1 and S2 in Supplementary Material could be misleading, (e.g., Membrane lipid composition).
First of all, we would like to thank the reviewer for all the comments and suggestions. Next, we proceed to answer all the comments, point by point.
Response 1:
We apologize for the mistake; we did not adjust adequately the protocol of our collaborators to what was really done. They did not measure cholesterol levels for us. In the case of gangliosides, they were measured, but the levels were very low and inconsistent all through the samples, for this reason, we did not include them.
Point 2:
Can the Authors detail slightly the reasons for choosing the extraction methods used?
Response 2:
The protocol was developed in the laboratory of Dr. Toshiro Okazaki, who are particularly interested in describing quantitatively and qualitatively the lipidome, being mostly focused on the phospholipid and sphingolipid composition. According to their experience, methods as Bligh and Dyer might not be sufficient to extract efficiently all lipids including sphingolipids, phospholipids, and glycolipids. For this reason, they develop this method using butanol which yields better results in terms of the number of lipid classes and molecular species detected. More details on the analytical procedures might be found in:
Ogiso, M. Taniguchi, S. Araya, S. Aoki, L.O. Wardhani, Y. Yamashita, Y. Ueda, T. Okazaki, Comparative Analysis of Biological Sphingolipids with Glycerophospholipids and Diacylglycerol by LC-MS/MS, Metabolites, 4 (2014) 98-114. doi: 10.3390/metabo4010098
Point 3:
In the Discussion, it would be interesting to read something about the expectations regarding the applicability of the results presented in clinical practice (e.g., possibilities of automation, cost-benefits, where does it stand among the other available tools – advantages/complementarities, etc.) and whether protein level and/or mRNA levels of the enzymes in PE-P metabolism could be more advantageous in comparison to the lipidomic analysis.
Response 3:
We have included a new paragraph in the discussion regarding the applicability of the results presented in clinical practice (see the last paragraph in the Conclusion section).
Regarding whether protein level and/or mRNA levels of the enzymes in PE-P metabolism could be more advantageous in comparison to the lipidomic analysis, it will depend on the exact nature of the alteration. That is if there is a global change (increase or decrease) in plasmalogens, and the enzyme accounting protein/gene is known, most probably is better to work with nucleic acid material, as it can be amplified. But if the alteration affects specifically to the profile at the molecular species level, then maybe it is more advantageous to perform a lipid analysis. The reality is that the hospital levels, lipidomic analysis describing complete profiles are still not used in the daily routine.
Sentences deserving revision/clarification:
Point 4:
L-61-64 “In this sense, our previous study analyzing the changes in the lipid signature along the colon…”
Response 4:
We have corrected this sentence. Now it reads:
“In this sense, our previous study analyzing the changes in the lipid signature along the colon epithelium demonstrates that there is a strict regulation at the molecular species level, concomitant to the colonocyte differentiation process and that this regulation, is clearly altered in the malignant tissue [8,15].”
Point 5:
L.89 “Altogether, these results verify the capacity of lipid profiles, whether from cells or from EVs, for 89 perceiving physiological alterations, such as those occurring during tissue malignization, and 90 consequently, in providing potential lipid biomarkers.”
Maybe change to “These results allow to verify … and to perceive…”
Response 5:
We have modified the sentence. Now it reads:
“Altogether, these results demonstrate the capacity of lipid profiles, whether from cells or EVs, for sensing a wide range of physiological alterations and, consequently, in providing potential lipid biomarkers.”
Point 6:
L167 “within the fatty acids found at the sn-2 fatty acid”. At the sn-2 position?
Response 6:
Yes, the reviewer is correct. We have corrected the mistake.
Point 7:
L168 “while 38:3 decreased to similar to the in situ cancer cells”
Response 7:
We have corrected the mistake. Now, it reads:
“while 38:3 were similar to the in situ cancer cells (3.0%).”
Point 8:
L174 “Consistent with previous reports40”
Response 8:
We have corrected this sentence and inserted the reference.
Point 9:
L256 “Interestingly, FAR1 and FAR2 were overexpressed in all cancer cells compared to primary cells, interesting at rather similar levels for each cancer cell ty” Please correct typos.
Response 9:
We have corrected the mistakes. Now, it reads:
“Interestingly, FAR1 and FAR2 were overexpressed in all cancer cells compared to primary cells, at rather similar levels for each cancer cell type”.
Point 10:
L285 “Just as with the cell lipidome, EV lipid analysis was sensitive enough to separate them according to their cell origin”
The sentence does not read as one in the beginning of a sub-section
Response 10:
We have rephrased the sentence.
Point 11:
L289 “Primary cells were separated from the rest due to high PE plasmalogen levels and low PC levels”
Please rephrase sentence for clarity
Response 11:
We have rephrased the sentence. Now, it reads:
“Thus, PCA results indicated that primary cells were segregated from the rest due to high PE plasmalogen levels and low PC levels (Figure 6B, C)”.
Point 12:
L312 “The latter together with the high PC and SM levels would imply that in situ-EV’s would lead to more rigid membranes.”
Please clarify. This comparison is made between EV and plasma membrane of Colo201 cells?
Response 12:
No, it is just comparing the EV lipid composition, in terms of lipid classes (Fig 6C). We have rephrased this and the previous sentences. Now, it reads:
“In terms of membrane lipid classes, EVs derived from in situ cancer cells were less diverse than those from the primary and Colo 201 cells. This homogeneity, together with the high PC and SM levels, would lead to more rigid membranes of the in situ derived EVs”.
Point 13:
L322 “Next, we investigated the differences in membrane lipid composition cell and EV lipidome”
Response 13:
We have corrected the mistake. Now, it reads:
“Next, we investigated the differences in cell and EV membrane lipidome by comparing their relative levels (Figure 6D).”
Point 14:
L360 “As in membrane classes, we compared line how a particular molecular species”
Response 14:
We have corrected the mistake. Now, it reads:
“As in membrane classes, we compared how a particular molecular species was distributed between cells and EVs within each cell line (Supplementary Figure 6)”.
Point 15:
L398 “whereas in liver cancer64”
Response 15:
The reference has been correctly inserted.
Point 16:
L442: “C18:0 saturated fatty acids69”
Response 16:
The reference has been correctly inserted.
Point 17:
L445-447 “Considering that apoptosis or exposure to noxious cues induces translocation to the outer leaflet of both diacyl and ether ethanolamine glycerophospholipids [56–58], it is possible that in cancer cells, the unusual translocation of PE plasmalogens could wrongly convey “a low plasmalogen level” signal, leading to overexpressed FAR1 levels, as reported in this study (Figure 5). “
What do the authors mean with unusual translocation – there is evidence for exposure of PE-P in the outer leaflet in cancer cells? Also, it seems to be in contradiction with the mechanism proposed in Figure 8. Please clarify.
Response 17:
We have carefully reviewed the literature and the reviewer is right, so far there has not been full confirmation that PE-P translocates to the outer leaflet in response to external or internal cues. However, it should be taken into account that the experiments where the translocation of diacyl-PE was shown, were done using duramycin (Stafford et al, Marconescu, et al), which also detects PE plasmalogens (Iwamoto et al). Hence, it cannot be ruled out that part of the translocated PE detected in these studies could contain PE plasmalogens.
Marconescu, A.; Thorpe, P.E. Coincident exposure of phosphatidylethanolamine and anionic phospholipids on the surface of irradiated cells. Biochim. Biophys. Acta - Biomembr. 2008, 1778, 2217–2224.
“PE was detected using a small antibiotic peptide, duramycin (from S. cinnamoneus) belonging to a family of tetracyclic polypeptides”.
Stafford, J.H.; Thorpe, P.E. Increased Exposure of Phosphatidylethanolamine on the Surface of Tumor Vascular Endothelium. Neoplasia 2011, 13, 299-IN2.
“In this report, we used duramycin to show that PE becomes specifically exposed on the surface of tumor EC and that treatment of cultivated EC with known tumor-associated stresses causes the formation of PE-positive blebs on the cell membrane”.
Iwamoto, T. Hayakawa, M. Murate, A. Makino, K. Ito, T. Fujisawa, T. Kobayashi, Curvature-dependent recognition of ethanolamine phospholipids by duramycin and cinnamycin, Biophys. J. 93 (2007) 1608–1619.
“The results indicate that both duramycin and cinnamycin selectively bind ethanolamine phospholipids, irrespective of whether they are of diacyl- or plasmalogen type”.
We have rephrased the text as follows:
“Using duramycin, a probe that binds both diacyl- or plasmalogen type [59], it was demonstrated that apoptosis or exposure to noxious cues induces translocation of ethanolamine glycerophospholipids [60–62]. Hence, it cannot be ruled out that the translocated PE could contain a fraction of PE plasmalogens. If so, this could trigger a “low plasmalogen level” signal at the inner leaflet [63] and leading to the overexpression of FAR1 levels (Figure 5)”.
Finally, we consider that this concept does not necessarily contradict the model depicted in Figure 8, as we consider that not all the newly synthesized PE-P has necessarily to translocate.